# Regulation of the glucocorticoid receptor via a BET-dependent enhancer drives antiandrogen resistance in prostate cancer

Neel Shah[1,2], Ping Wang[3], John Wongvipat[1], Wouter R Karthaus[1], Wassim Abida[4], Joshua Armenia[1], Shira Rockowitz[3], Yotam Drier[5], Bradley E Bernstein[5], Henry W Long[6], Matthew L Freedman[6], Vivek K Arora[7], Deyou Zheng[3], Charles L Sawyers[1,8]*

[1]Human Oncology and Pathogenesis Program, Memorial Sloan Kettering Cancer Center, New York, United States; [2]The Louis V. Gerstner Graduate School of Biomedical Sciences, Sloan Kettering Institute, Memorial Sloan Kettering Cancer Center, New York, United States; [3]Department of Neurology, Genetics and Neuroscience, Albert Einstein College of Medicine, Bronx, United States; [4]Department of Medicine, Memorial Sloan Kettering Cancer Center, New York, United States; [5]Department of Pathology, Massachusetts General Hospital and Harvard Medical School, Boston, United States; [6]Department of Medical Oncology, Dana-Farber Cancer Institute and Harvard Medical School, Boston, United States; [7]Division of Medical Oncology, Washington University School of Medicine, St Louis, United States; [8]Howard Hughes Medical Institute, Memorial Sloan Kettering Cancer Center, New York, United States

**Abstract** In prostate cancer, resistance to the antiandrogen enzalutamide (Enz) can occur through bypass of androgen receptor (AR) blockade by the glucocorticoid receptor (GR). In contrast to fixed genomic alterations, here we show that GR-mediated antiandrogen resistance is adaptive and reversible due to regulation of GR expression by a tissue-specific enhancer. GR expression is silenced in prostate cancer by a combination of AR binding and EZH2-mediated repression at the GR locus, but is restored in advanced prostate cancers upon reversion of both repressive signals. Remarkably, BET bromodomain inhibition resensitizes drug-resistant tumors to Enz by selectively impairing the GR signaling axis via this enhancer. In addition to revealing an underlying molecular mechanism of GR-driven drug resistance, these data suggest that inhibitors of broadly active chromatin-readers could have utility in nuanced clinical contexts of acquired drug resistance with a more favorable therapeutic index.
DOI: https://doi.org/10.7554/eLife.27861.001

*For correspondence: sawyersc@mskcc.org

## Introduction

Drugs targeting the androgen receptor (AR) signaling pathway form the backbone of therapy for advanced prostate cancer. Despite initial responses to androgen deprivation therapy (ADT), patients with metastatic disease invariably progress to a stage termed castration-resistant prostate cancer (CRPC). Overall survival of CRPC patients is significantly improved by treatment with next generation AR pathway inhibitors such as abiraterone (Abi) and enzalutamide (Enz) that more effectively target

the AR signaling axis (*Scher et al., 2012*; *de Bono et al., 2011*). However, acquired resistance to these drugs is a major clinical problem.

A common mechanism of acquired resistance to all targeted therapies is genetic mutation. This is best exemplified by kinase inhibitors, where relapse occurs through clonal expansion of tumor cells harboring a specific, irreversible mutation in the relevant kinase target (*Gorre et al., 2002*; *Shah et al., 2002*; *Pao et al., 2005*). The same is true for nuclear receptor antagonists. For example, AR mutations or gene amplifications can confer resistance to antiandrogen therapy in prostate cancer, including Enz (*Balbas et al., 2013*; *Joseph et al., 2013*; *Watson et al., 2015*), and estrogen receptor (ER) mutations confer resistance to ER pathway therapies in breast cancer (*Robinson et al., 2013*; *Toy et al., 2013*; *Jeselsohn et al., 2014*).

Resistance to targeted therapies can also arise due to reversible mechanisms that are not mutation based. BRAF mutant colorectal cancers are resistant to kinase inhibition due to activation of EGFR caused by a BRAF-dependent negative feedback loop (*Prahallad et al., 2012*). Similarly, Notch-mutant T-cell acute lymphoblastic leukemias (T-ALL) treated with gamma-secretase inhibitors (GSIs) develop an epigenetic, reversible state and subsequent activation of downstream transcriptional programs that bypass the GSI blockade (*Knoechel et al., 2014*). These leukemias can be targeted via the use of BET inhibitors in combination with GSIs, establishing a basis for incorporating epigenetic modulators in combination with targeted therapies to overcome resistance.

We and others have previously reported the role of GR upregulation in conferring resistance to Enz (*Isikbay et al., 2014*; *Li et al., 2017*; *Arora et al., 2013*). GR can bind to and drive the expression of a subset of AR target genes, thus allowing the cancer to progress despite ongoing AR blockade. Here we show that GR expression in this context is partially reversible upon withdrawal of Enz and identify a previously unknown enhancer present in normal and malignant prostate tissue that regulates GR expression. We also reveal a pharmacologic strategy to target GR expression in Enz-resistant tumors with surprising selectivity using BET inhibitors.

## Results

### A tissue-specific enhancer regulates GR expression in CRPC

To study acquired resistance mechanisms to ADT, we employed the AR-dependent LNCaP/AR (LNAR) mouse xenograft model, previously used to demonstrate the activity of Enz (*Tran et al., 2009*), and we identified GR upregulation as a driver of Enz resistance (*Arora et al., 2013*). LNAR cells were injected subcutaneously into castrate mice and treated with either vehicle or Enz for an extended period of time until resistant tumors formed. Vehicle-treated (LNAR') and Enz-resistant (LREX') tumors were then adapted back into in vitro culture conditions for further molecular characterization (*Figure 1A*). GR upregulation was found in 8 of 13 independent Enz-resistant tumor clones. GR expression in LREX' cells was dynamic and could be toggled on and off based on Enz exposure, consistent with an AR-mediated negative feedback (*Figure 1A*, *Figure 1—figure supplement 1A*). Conversely, Enz treatment did not result in significant GR upregulation in Enz-sensitive LNAR' cells or in GR-negative resistant clones (*Figure 1—figure supplement 1A*), suggesting a second level of GR regulation beyond AR inhibition. The importance of GR as a resistance mechanism was also evident in the CRPC organoid line (MSK-PCa2) (*Gao et al., 2014*), where dexamethasone (Dex) treatment partially restored growth and rescued AR target gene expression in the presence of Enz (*Figure 1—figure supplement 1B,C*).

Whole exome sequencing of LNAR' and LREX' cells did not reveal any gene-coding mutations associated with Enz resistance; therefore, we considered epigenetic mechanisms. Previous studies of GR transcriptional regulation have largely focused on the proximal promoter, which is the primary mechanism of GR regulation in most tissue types (*Breslin et al., 2001*; *McGowan et al., 2009*; *Nobukuni et al., 1995*; *Nunez and Vedeckis, 2002*). To examine GR regulation in the context of this prostate model, we used chromatin-immunoprecipitation followed by sequencing (ChIP-seq) to survey the chromatin landscape for the H3K4me3, H3K4me1 and H3K27ac histone modifications, then examined these marks across the GR gene (*NR3C1*) locus (*Figure 1B*). This analysis identified a clear enhancer (revealed by the H3K4me1 track) (*Rada-Iglesias et al., 2011*; *Creyghton et al., 2010*) in LNAR' and LREX' cells, regardless of Enz treatment. Consistent with the high level of GR mRNA expression in LREX' cells treated with Enz, this enhancer and the GR promoter both had

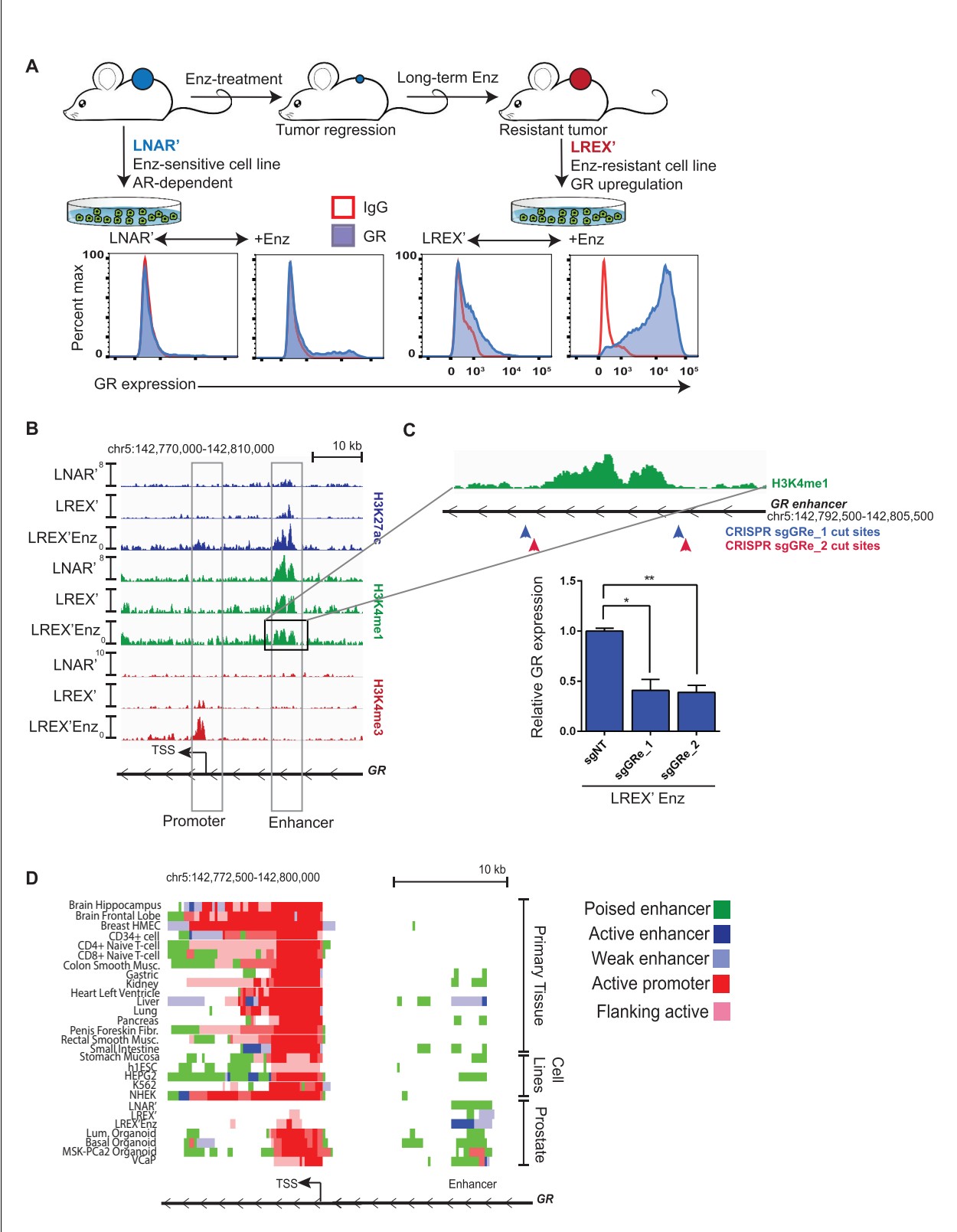

**Figure 1.** Characterization of a tissue-specific GR (NR3C1) enhancer in prostate cells and Enz-resistant prostate cancer. (**A**) Top - LNAR' Enz-sensitive and LREX' Enz-resistant cell lines were derived from in vivo xenograft experiments. Bottom - Flow cytometry on LNAR' and LREX' cell lines for GR expression in the presence and absence of Enz (1 uM). (**B**) ChIP-seq tracks for H3K27ac (blue), H3K4me1 (green), H3K4me3 (red) at the GR (NR3C1) locus for LNAR', LREX' and LREX'Enz cell lines define the promoter and enhancer regions. Normalized ChIP-seq read counts at GR enhancer/promoter

*Figure 1 continued on next page*

*Figure 1 continued*

region (LNAR', LREX', LREX'Enz; **=Z score>2): H3K27ac (20.86, 30.57, 72.46**); H3K4me1 (59.62, 36.69, 42.61); H3K4me3 (7.97, 24.54, 94.91**). (C) Top - Experimental design depicting two CRISPR guide pairs (sgGRe_1 and sgGRe_2) flanking the GR (NR3C1) enhancer to excise the entire enhancer region. Bottom - RT-qPCR for GR expression from sorted cells transfected with two different CRISPR guide pairs (sgGRe_1 and sgGRe_2) and a non-targeting (sgNT) control. (D) ChromHMM prediction of promoter and enhancer regulatory elements at the GR (NR3C1) locus using ChIP-seq data for H3K27ac, H3K4me1, H3K4me3 across multiple tissue types (from Roadmap Epigenomics Project), cell lines (from ENCODE database), and prostate cell and organoid lines.

DOI: https://doi.org/10.7554/eLife.27861.002

The following figure supplement is available for figure 1:

**Figure supplement 1.** Further characterization of GR expression and regulation in prostate cell and organoid models.

DOI: https://doi.org/10.7554/eLife.27861.003

clearly defined active chromatin marks (H3K27ac and H3K4me3, respectively). Conversely, the lack of these active marks in LNAR' cells signified a poised GR enhancer state.

Curiously, this enhancer has not been reported in earlier studies of GR transcriptional control, largely conducted in other tissue types. To investigate the role of the enhancer in regulating GR expression in this prostate cancer model, we designed two independent CRISPR guide RNA pairs (sgRNA) flanking the ~4.7 kb enhancer to excise the entire region (*Figure 1C*). Anticipating that LREX' cells treated with Enz may not tolerate sustained silencing of GR expression, we deleted the enhancer using a transient approach. LREX' cells were transiently transfected with the guide pairs and a GFP-linked sgRNA-Cas9 vector, then sorted based on GFP expression. PCR analysis of DNA isolated from the GFP-positive population, followed by sequencing, confirmed successful excision of the entire enhancer (*Figure 1—figure supplement 1D,E*). LREX' cells lacking the enhancer showed a 60 percent reduction in Enz-induced GR expression as measured by quantitative PCR (qPCR) (*Figure 1C*), validating its importance in regulating GR transcription.

Having confirmed the importance of the enhancer in this model, we searched the Roadmap Epigenomics Project and ENCODE databases, together with ChromHMM software which integrates multiple histone modifications (*Ernst and Kellis, 2012*), for evidence of its existence in other tissue types (*Figure 1D*, *Figure 1—figure supplement 1G*). This analysis confirmed active promoters at the GR locus in most tissues, but minimal evidence of a GR enhancer. Exceptions include kidney and liver tissue (and HEPG2 liver cancer cells), with evidence of a weak or poised enhancer, and pancreas and gastric tissue with spotty chromatin marks in this region. One limitation of the Roadmap Epigenomics Project is the lack of any prostate tissue or prostate cell line data. Therefore, we expanded our ChIP-seq analysis of chromatin marks beyond the LNAR'/LREX' model to include additional prostate samples. Due to challenges in obtaining high quality ChIP-seq data from primary prostate tissue, we used organoid cultures to propagate normal and malignant human prostate specimens (*Gao et al., 2014*; *Karthaus et al., 2014*) (*Figure 1—figure supplement 1F*). This allowed us to generate high quality ChIP-seq data from two normal human prostate samples (basal and luminal cells) and from the human CRPC organoid line MSK-PCa2 (*Gao et al., 2014*). We also included publically available ChIP-seq data from the VCaP prostate cancer cell line (*Yu et al., 2010*). In contrast to the non-prostate tissues, this analysis revealed a clearly defined enhancer (H3K4me1) in every prostate tissue examined, including in Enz-treated LREX' cells which also have the active H3K27ac enhancer mark (*Figure 1D*). Taken together, these data reveal an enhancer at the GR locus in normal and malignant prostate cancer cells, not present in most other tissues, that plays a critical role in regulating GR mRNA expression in the Enz-resistant state.

## GR expression in prostate cancer is regulated via AR occupancy at the upstream enhancer and polycomb-mediated silencing

Previous studies have reported GR expression in normal prostate tissue but not in primary prostate cancer (*Yemelyanov et al., 2007*; *Mohler et al., 1996*), which we confirmed by immunohistochemistry (IHC) in five tumor-normal pairs. GR is robustly expressed in both basal and luminal cells in normal prostate tissue, but substantially reduced in primary prostate cancer (*Figure 2—figure supplement 1A*). To examine GR expression across the entire spectrum of prostate cancer from primary to late stage metastatic disease, we conducted a meta-analysis of several large-scale RNA expression datasets (*Grasso et al., 2012*; *Robinson et al., 2015*; *Cancer Genome Atlas Research*

*Network et al., 2013*). Consistent with the IHC data, we observed a significant decrease in GR mRNA levels in primary prostate cancer relative to normal tissue, with a further decrease in CRPC samples from patients who have not received Enz or Abi (*Figure 2A*). However, mean GR mRNA levels were significantly increased in CRPC samples obtained after Enz/Abi treatment but not back to the levels seen in primary disease. More detailed analysis of the post-Enz/Abi patients revealed that the increase in mean GR level is explained, in part, by a subset of patients with higher GR expression, consistent with our previous finding of increased GR protein expression in a subset of Enz-treated patients (*Arora et al., 2013*). We also performed computational analyses to compare tumor and stromal cell content and found no differences in the CRPC pre- versus post-Enz/Abi samples (*Figure 2—figure supplement 1A*).

Next we examined AR mutations, AR splice variants and total AR mRNA levels in Enz/Abi-resistant CRPC tumors and their relationship with GR expression. 7 of 50 post-Enz/Abi samples had detectable AR mutations (all with AR L702H, known to be activated by prednisone and enriched in post-Abi patients) with no correlation with GR levels (*Figure 2—figure supplement 1D*). The AR-V7 splice variant was negatively correlated with GR expression (Spearman R = −0.39, p=0.03) (*Figure 2—figure supplement 1E*), suggesting that GR and AR-V7 may define mutually exclusive subsets of Enz/Abi-resistant patients, although larger cohorts are needed for more definitive analysis (*Antonarakis et al., 2014*; *Scher et al., 2016*). There was with no significant increase in AR mRNA expression in the pre- versus post-Enz/Abi cases (*Figure 2—figure supplement 1C*). In summary, this analysis reveals a loss of GR mRNA levels during the transition from normal prostate to prostate cancer, and a subsequent increase in GR expression in the post-Enz/Abi treatment setting.

To explore the mechanism underlying this dynamic regulation of GR expression in prostate cancer progression, we first returned to the LNAR'/LREX' model system where continuous Enz exposure is required to maintain high levels of GR expression (*Figure 1A*). We previously reported that dihydrotestosterone (DHT) treatment suppresses GR expression in LREX cells (*Arora et al., 2013*) and noted an AR binding peak that, interestingly, maps precisely to this newly defined GR enhancer, adjacent to FOXA1 and HOXB13 motifs (*Figure 2—figure supplement 2A*). We confirmed Enz-reversible AR binding at this site in both LNAR' and LREX' cells by chromatin immunoprecipitation followed by polymerase-chain reaction (ChIP-PCR) (*Figure 2B*). To address the question of how GR expression is silenced in localized prostate cancer, we turned to a set of matched tumor/normal primary clinical samples whose AR ChIP-seq profiles were recently characterized (*Pomerantz et al., 2015*). Remarkably, AR binding peaks were detected at the GR enhancer in 5 of 5 tumors, but not in the matched normal tissue (*Figure 2C*), precisely correlating with reduced GR mRNA levels seen in primary cancers versus normal tissue (*Figure 2A*). We therefore postulate that AR binding at the GR enhancer site is repressive, as has been reported for an AR binding site within the AR genomic locus (*Cai et al., 2014*). The differential AR binding at this enhancer in normal versus cancer, despite comparable levels of AR expression, is consistent with extensive reprogramming of the AR cistrome reported in human prostate tumors (*Pomerantz et al., 2015*).

While repressive AR binding is sufficient to explain reduced GR expression in primary cancer, we postulated that other mechanisms may contribute to the further decline in GR levels observed in CRPC patients. Many promoters and enhancers are regulated through the repressive histone mark H3K27me3 (*Rada-Iglesias et al., 2011*; *Creyghton et al., 2010*; *Zentner et al., 2011*), which is deposited on histones via the EZH2 enzyme, part of the polycomb repressive complex 2 (PRC2). Due to technical challenges in obtaining high quality H3K27me3 ChIP-seq data from primary prostate tissue, we again utilized human prostate organoid cultures to gather data from normal prostate cells and an additional CRPC sample (MSK-PCa2) as well as the isogenic LNAR'/LREX' resistance model. We observed increased H3K27me3 at the GR promoter and enhancer in the cancer samples (MSK-PCa2, LNAR') compared to normal prostate cells, but loss of the H3K27me3 mark in Enz-resistant LREX' cells (*Figure 2D*). The results show that H3K27me3 at the GR locus is correlated with reduced expression. To determine if the H3K27me3 mark is responsible for GR repression, we treated LNAR' cells with the EZH2 inhibitor GSK126 to erase the repressive mark (*Figure 2—figure supplement 2B*), then measured GR levels in the presence or absence of Enz. Although treatment with GSK126 alone had no effect, GR levels rose more than 30-fold in combination with Enz, to levels comparable to those seen in LREX' cells treated with Enz alone (*Figure 2E*). We observed similar cooperativity between GSK126 and Enz in MSK-PCa2 tumor organoids but not in normal prostate organoids that lack H3K27me3 or AR binding at the GR locus (*Figure 2—figure supplement 2C*).

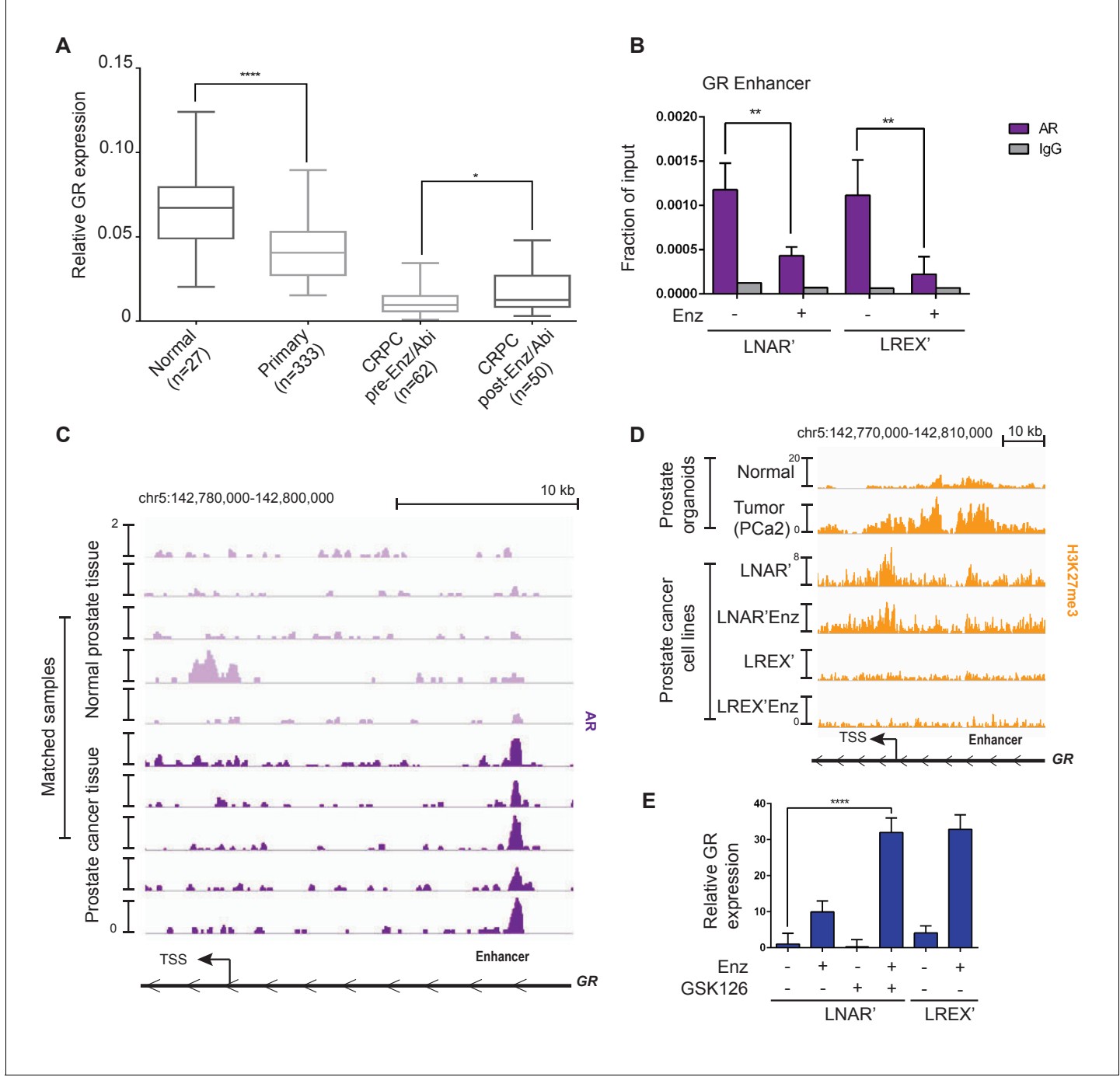

**Figure 2.** Regulation of GR expression in prostate cancer disease progression at the GR (NR3C1) locus via AR binding and polycomb-mediated H3K27me3 repression. (A) Combination of TCGA (normal prostate [n = 27] and primary prostate cancer [n = 333]) and SU2C (CRPC pre-Enz/Abi [n = 62] and CRPC post-Enz/Abi [n = 50]) gene expression datasets showing GR expression in prostate cancer disease progression. GR levels across datasets normalized to UBC housekeeping gene. Boxplot shows median GR expression with 5–95 percentile bars. (B) ChIP-qPCR for AR at the GR (NR3C1) enhancer for LNAR' and LREX' cell lines maintained on and off Enz. (C) ChIP-seq tracks for AR at the GR (NR3C1) locus in patient-matched normal prostate tissue and primary prostate cancer tissue samples. (D) ChIP-seq tracks for H3K27me3 at the GR (NR3C1) locus in normal luminal prostate organoids and advanced prostate cancer (MSK-PCa2) organoids (top), and LNAR' and LREX' cell lines maintained on and off Enz (1 uM) (bottom). Normalized ChIP-seq read counts at GR enhancer/promoter region (LNAR', LNAR'Enz, LREX', LREX'Enz; **=Z score>2): H3K27me3 (65.44**, 56.99**, 17.97, 8.53). (E) RT-qPCR for GR expression in LNAR' cells treated for 2 weeks with different combinations of Enz (1 uM) and GSK126 (3 uM), and LREX' cells treated with and without Enz (1 uM).

DOI: https://doi.org/10.7554/eLife.27861.004

*Figure 2 continued on next page*

*Figure 2 continued*

The following figure supplements are available for figure 2:

**Figure supplement 1.** Analysis of GR expression in multiple datasets and primary patient samples.

DOI: https://doi.org/10.7554/eLife.27861.005

**Figure supplement 2.** Regulation of GR expression at the GR (NR3C1) locus.

DOI: https://doi.org/10.7554/eLife.27861.006

Together, these data suggest that GR expression in prostate tissue is silenced during the transition to cancer by a combination of AR binding and PRC2-mediated repression at the GR locus. We postulate that the PRC2 mark can be erased in late stage CRPC (post-Enz/Abi), which enables cells to express GR when exposed to Enz, as Enz treatment displaces AR repression from the GR enhancer.

## BET inhibition impairs GR expression and resensitizes tumors to Enz

Having identified an enhancer responsible for GR upregulation in Enz-resistant CRPC that is marked by H3K27ac, we asked if BET family proteins, which bind acetylated lysine motifs at enhancers and help drive the expression of key tissue-specific genes (*Dhalluin et al., 1999*; *Filippakopoulos et al., 2012*; *Jang et al., 2005*; *Shi and Vakoc, 2014*), regulate GR expression in this setting. Indeed, in vitro treatment of LREX′ cells with the BET inhibitor JQ1 (*Filippakopoulos et al., 2010*) resulted in a dose-dependent decrease of GR expression (*Figure 3A*). Consistent with BET bromodomains being epigenetic readers as opposed to writers or erasers, we observed no changes in H3K27ac, H3K27me3 or H3K4me1 at the GR enhancer with JQ1 treatment (*Figure 3—figure supplement 1A*). BET inhibitors are reported to interfere with AR function, but through a different mechanism, by blocking recruitment of AR to chromatin and impairing expression of downstream target genes rather than directly lowering AR mRNA levels (*Asangani et al., 2014*; *Faivre et al., 2017*; *Chan et al., 2015*); however, we did not observe effects of JQ1 on AR binding at the GR enhancer (*Figure 3—figure supplement 1B*).

To distinguish between these effects of BET inhibition on GR versus AR (and to determine the impact on Enz-resistance in a GR-driven model), we treated isogenic LNAR′ and LREX′ xenografts in parallel with vehicle, JQ1, Enz or JQ1+Enz, and performed RNA-seq on the tumors to evaluate effects on gene expression (*Figure 3B,C*). Remarkably, LNAR′ tumors showed no response to JQ1 treatment despite retaining their well-documented sensitivity to Enz and dependency on AR-signaling (*Figure 3B*). This is particularly striking because RNA-seq analysis of JQ1-treated LNAR′ tumors confirmed that the canonical BET-dependent gene *MYC* was potently downregulated, thereby validating sufficient JQ1 exposure, with no changes in AR target genes such as *NKX3.1* (*Figure 3C*) or an AR target gene signature, as discussed below (*Figure 3E*). In contrast to LNAR′ tumors, LREX′ tumors were resistant to Enz treatment, as expected. However, LREX′ tumors regressed when both JQ1 +Enz were delivered in combination (*Figure 3B*), effectively re-sensitizing resistant tumors to Enz. Of note, some of the most potently downregulated genes in JQ1-treated LREX′ tumors were *GR (NR3C1)* itself, the GR target gene *SGK1*, and *MYC*, whereas *NKX3.1* was largely unchanged (*Figure 3C*).

To explore potential reasons for our failure to detect an effect of JQ1 on AR target genes in vivo, we treated LREX′ cells with a range of doses (0.01–1 uM) in vitro and measured impact on several AR versus GR target genes, selected based on previously reported AR-biased versus GR-biased gene signatures (*Arora et al., 2013*). The AR target genes *NKX3.1* and *TMPRSS2* were upregulated at lower JQ1 concentrations (10 nM) but moderately suppressed at higher concentrations (1 uM) (*Figure 3D*). In contrast, the GR target genes *SGK1* and *FKBP5*, as well as *GR (NR3C1)* itself and *MYC*, were consistently and more profoundly inhibited in a dose-dependent fashion. To further distinguish between the effects of JQ1 on AR versus GR signaling, we performed gene set enrichment analysis (GSEA) on the in vivo LREX′ tumors. This analysis revealed that Enz primarily inhibits AR-biased genes in LREX′ xenografts, whereas JQ1 primarily inhibits GR-biased genes (*Figure 3E*). It is not until tumors are treated with a combination of JQ1+Enz together do we see inhibition of both AR and GR signaling pathways (*Figure 3E*), as well as LREX′ tumor regression (*Figure 3B*). The selective effect of JQ1 on GR but not AR in this model is likely explained by the dose-dependent

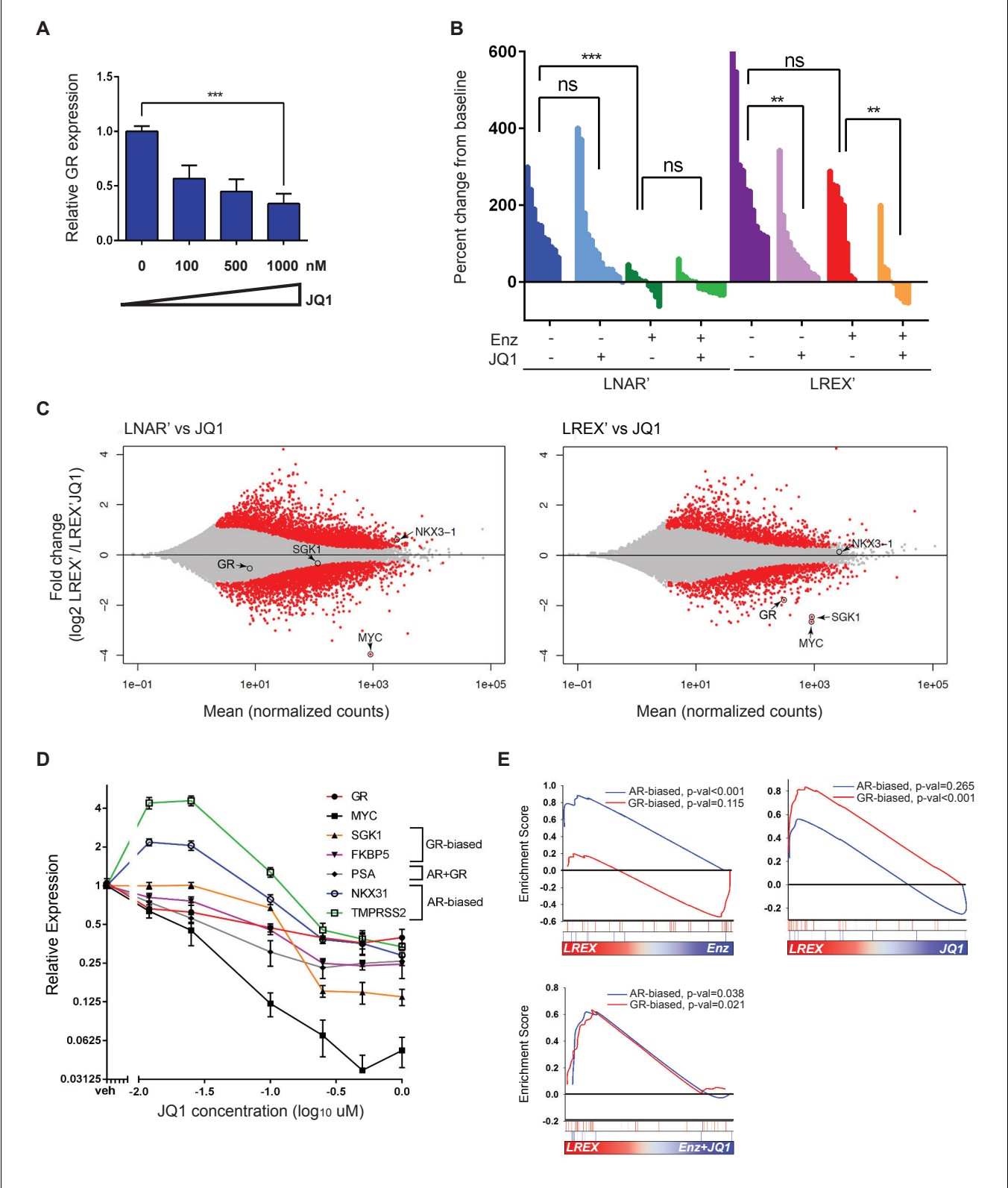

**Figure 3.** Sensitivity of LREX' resistant tumors to BET inhibition by modulation of GR expression. (**A**) RT-qPCR for GR expression in LREX'Enz cells treated in vitro with increasing doses of JQ1 for 24 hr. (**B**) Waterfall plot showing in vivo LNAR' and LREX' xenograft tumor growth, treated with different combinations of Enz (10 mg/kg daily) and JQ1(50 mg/kg daily) for 4 weeks after tumors are established. (**C**) MA plot showing differential gene expression analysis from RNA-seq data of in vivo LNAR' (left) and LREX' tumors (right) treated with JQ1 (50 mg/kg) after 4 weeks of treatment. Red

*Figure 3 continued on next page*

Figure 3 continued

points are differentially expressed genes (DEGs) that have an adjusted p-val <0.05. MYC, GR, NKX3.1 (AR target) and SGK1 (GR target) genes are shown on plot. (D) JQ1 treatment dose-response curve in LREX'Enz cells for 48 hr, measuring gene expression changes normalized to veh treatment for GR (NR3C1), MYC, SGK1, FKBP5, PSA (KLK3), NKX3.1, TMPRSS2. Error bars represent standard error. (E) Gene set enrichment analysis (GSEA) using AR-biased and GR-biased gene sets on LREX' tumors treated with Enz (10 mg/kg), JQ1(50 mg/kg), or a combination of Enz +JQ1 after 4 weeks of treatment.

DOI: https://doi.org/10.7554/eLife.27861.007

The following figure supplement is available for figure 3:

**Figure supplement 1.** Effects of JQ1 treatment on histone marks at GR (NR3C1) locus.

DOI: https://doi.org/10.7554/eLife.27861.008

sensitivity of their downstream targets (revealed in vitro) coupled with well documented challenges in achieving sustained JQ1 exposure in vivo (*Matzuk et al., 2012*).

To explore the mechanism underlying the relative selectivity of BET inhibition on GR versus AR target genes, we utilized a technique called Chem-seq to map JQ1 interactions with chromatin across the genome using a biotinylated version of the small molecule (bio-JQ1) (*Anders et al., 2014*). This method has the additional advantage of surveying all JQ1 target proteins at once rather than limiting our analysis to a single BET-bromodomain containing family member (*Filippakopoulos et al., 2010*). As expected, we observed strong bio-JQ1 binding at the MYC gene locus in LNAR' and LREX' cells in the presence or absence of Enz. The specificity of the bio-JQ1 signal was confirmed by BRD4 ChIP-seq which showed binding of the well-known JQ1 target BRD4 at precisely the same locus (*Figure 4A*) as well as by the highly significant correlation of bio-JQ1 with H3K27ac (*Figure 4—figure supplement 1A*). We also observed significant bio-JQ1 binding at the GR enhancer, but only in LREX' cells treated with Enz, when the H3K27ac mark is present (*Figure 4B*). Interestingly, we did not detect BRD4 binding at the enhancer, suggesting that a different JQ1 target protein is likely responsible for driving GR expression.

To address the question of how JQ1 might have relatively selective effects on GR, we conducted an unbiased genome-wide analysis of bio-JQ1 binding in LNAR' and LREX' cells. Remarkably, the GR enhancer is amongst the most differential binding sites, showing strong preferential binding in LREX' cells, whereas binding at the MYC locus is unchanged (*Figure 4C*). This enriched binding at the GR locus, together with the primary transcriptional effects on GR and not AR targets, suggests that the anti-tumor activity of JQ1+Enz in the LREX' xenograft model is most likely through GR inhibition. To test this hypothesis, we introduced a doxycycline (Dox)-inducible allele of GR under the control of a BET-independent promoter into LREX' cells and confirmed robust Dox-inducible GR (and downstream SGK1) expression in tumors despite JQ1 treatment (*Figure 4—figure supplement 1B,C*). Remarkably, GR overexpression in the setting of combined JQ1+Enz rescued tumor growth (*Figure 4D*), thereby demonstrating that the anti-tumor activity of BET inhibition in this model is mediated by GR suppression.

## Discussion

In summary, this report provides new mechanistic insight into how GR expression is restored in a subset of CRPC patients with acquired resistance to Enz, and how BET inhibitors can have surprisingly selective effects on gene expression in specific contexts. Both findings have implications for the future clinical management of CRPC and for the broader topic of reversible and epigenetic mechanisms of acquired drug resistance in cancer.

Based on a careful examination of GR levels across the full clinical spectrum of prostate cancer progression and in several model systems, we find that GR expression in normal prostate epithelial cells is silenced through a two-step process during the transition to malignancy (*Figure 4E*). First, we have characterized a novel enhancer at the GR locus, demonstrated that this enhancer is required for GR expression, and shown that AR binding at the enhancer is coupled with reduced GR expression. We propose that the acquisition of AR binding at the enhancer, which triggers a decline in GR expression, is likely a consequence of the reprogrammed AR cistrome in prostate cancer versus normal tissue, which is enriched at HOXB13 and FOXA1 sites (*Pomerantz et al., 2015*). Second, we have demonstrated increased levels of the repressive H3K27me3 mark across the GR promoter and

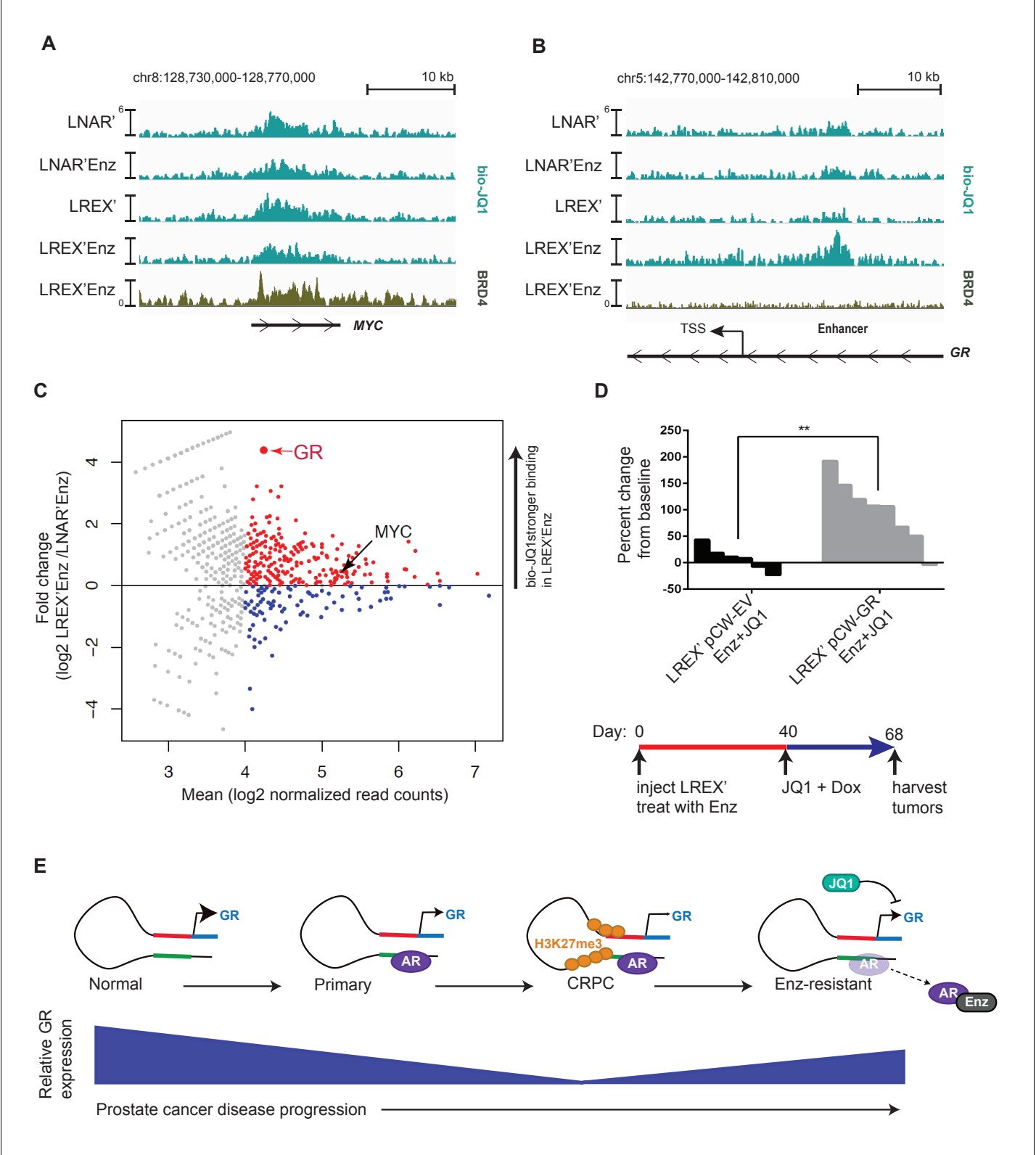

**Figure 4.** Mechanism of JQ1 drug action in LREX' resistant tumors. Chem-Seq tracks for bioJQ1 (light tracks) and ChIP-seq tracks for BRD4 (dark track) in LNAR' and LREX' cell lines treated with and without Enz (1 uM), showing the MYC locus (**A**) and GR gene (NR3C1) locus (**B**). Normalized Chem-seq/ChIP-seq read counts at MYC promoter region: (LNAR', LNAR'Enz, LREX', LREX'Enz; **=Z score>2): bio-JQ1 (36.53, 48.50, 37.10, 38.49); BRD4 (31.56**). GR enhancer: bio-JQ1 (45.98, 45.46, 33.85, 91.68**); BRD4 (16.49). (**C**) Differential peak analysis of bio-JQ1 Chem-Seq between LREX'Enz and LNAR'Enz.

*Figure 4 continued on next page*

*Figure 4 continued*

Colored points have mean log2 normalized reads >4; blue points are bio-JQ1 peaks higher in LNAR'Enz cells, red points are bio-JQ1 peaks higher in LREX'Enz cells. Differential GR and MYC peaks are shown on plot. (D) Top - Waterfall plot showing in vivo LREX' xenograft tumor growth with doxycycline-inducible GR over-expression (pCW-GR) or empty vector (pCW-EV) control. Tumors treated with Enz (10 mg/kg), JQ1 (50 mg/kg), and doxycycline for 4 weeks once tumors are established. Bottom - experimental design for GR overexpression rescue experiment over the course of 68 days. (E) Model for GR regulation in prostate cancer disease progression.
DOI: https://doi.org/10.7554/eLife.27861.009
The following figure supplement is available for figure 4:

**Figure supplement 1.** Further analysis of JQ1 anti-tumor activity and GR selectivity.
DOI: https://doi.org/10.7554/eLife.27861.010

enhancer in CRPC models, but not in normal prostate tissue organoids, again correlating with reduced GR expression. Increased levels of EZH2 expression observed in CRPC (*Varambally et al., 2002*) may play a role in the progressive accumulation of the repressive H3K27me3 mark. Finally, pharmacological EZH2 inhibition erases the repressive mark and partially restores GR expression, as long as Enz is also present to displace AR repression from the GR enhancer. Collectively, these findings demonstrate that acquired Enz resistance can occur in CRPC patients through reactivation of this normally silenced GR locus. The precise levels of GR re-expression required for Enz resistance will require further study, as well as whether discontinuation of Enz treatment results in a decline in GR levels in patient tumors.

A remaining question is whether the silencing of GR in primary prostate cancer has functional consequences in tumor progression. Prior work has suggested that GR can attenuate AR signaling and inhibit growth in AR-positive prostate cancer models, raising the possibility that tumor cells gain a growth advantage upon GR silencing (*Yemelyanov et al., 2007*; *Sahu et al., 2013*). However, in the setting of GR-driven Enz-resistance, where AR is still inhibited, GR can now function as an activator of growth by substituting for AR.

We also report a remarkably selective effect of BET inhibition in restoring sensitivity to Enz in CRPC by suppression of GR expression, which we demonstrate through a combination of RNA-seq, Chem-seq, and GR rescue experiments. This selectivity for GR inhibition (over AR inhibition) is surprising in light of considerable evidence that BET inhibitors can impair AR function (*Asangani et al., 2014*; *Faivre et al., 2017*; *Chan et al., 2015*). We postulate that the failure to see AR pathway inhibition in our models is due to a dose-dependent, selective inhibition of GR versus AR target genes. Further, differences in the chromatin context of the most BET-dependent genes (e.g. *GR (NR3C1)*, *MYC*) versus AR target genes (e.g. *NKX3.1, TMPRSS2*) may explain the differential binding of bio-JQ1 at these sites as seen by Chem-seq. Prior work, primarily in hematologic malignancies, showing that genes whose enhancers have high levels of H3K27ac (e.g. *MYC*) tend to be exquisitely sensitive to BET inhibition provides additional support to this hypothesis (*Shao et al., 2014*; *Mertz et al., 2011*; *Ott et al., 2012*; *Delmore et al., 2011*). Regardless of the mechanism, an important implication of this differential sensitivity is that the dose of BET inhibitor required for therapeutic impact is likely to vary depending on the clinical context. In patients with acquired Enz resistance driven by GR, our data predicts that clinical benefit may be seen at lower doses than those required for AR inhibition. This hypothesis should be testable using newer BET inhibitors with more favorable in vivo drug-like properties (relative to JQ1) that are currently in clinical development (*Faivre et al., 2017*; *Albrecht et al., 2016*; *Mirguet et al., 2013*; *Berthon et al., 2016*). The results could provide critical insight into designing future clinical trials to optimize therapeutic impact while avoiding the significant dose-limiting toxicities observed in current single agent BET inhibitor trials (*Kharfan-Dabaja, 2016*).

## Materials and methods

### Cell lines and organoid culture conditions

Cell lines and organoid lines were derived and cultured in conditions as previously described (*Arora et al., 2013*; *Gao et al., 2014*; *Karthaus et al., 2014*; *Drost et al., 2016*). LNAR' (previously known as CS1 (16)) and LREX' cell lines were derived from in vivo xenograft experiments, and

adapted back into in vitro culture conditions with or without 1 uM Enz. Organoid lines were derived from human normal luminal prostate tissue (CD26 +cells) or human advanced prostate cancer tissue (MSK-PCa2 organoid line), and maintained in 3D culture conditions, as previously described. All cell and organoid lines were authenticated by exome sequencing methods, and were negative for mycoplasma contamination testing.

## Intracellular GR staining and flow cytometry

Flow cytometry for GR in LNAR' and LREX' cell lines were performed as previously described (*Arora et al., 2013*). Cells were fixed and permeabilized using a fixation/permeabilization kit (eBioscience cat no. 00-5523-00), and incubated with either Rabbit (DA1E) mAb IgG XP Isotype control, or glucocorticoid receptor (D6H2L) XP Rabbit mAb (Cell Signaling Technology). Secondary antibody used was Allophycocyanin-AffiniPure F(ab) Fragment Donkey Anti-Rabbit IgG (Jackson ImmunoResearch Laboratories). Flow cytometry was performed on LSRII (BD Biosciences), and analysis was conducted using FlowJo software.

## Chromatin immunoprecipitation (ChIP) and sequencing analysis

ChIP experiments were performed as previously described (*Arora et al., 2013*), using SDS-based buffers. Antibodies were used at a concentration of 5 ug per 1 mL of IP buffer, which encompassed approximately 8 million cells per IP. Antibodies used were: H3K27ac (Abcam ab4729), H3K4me1 (Abcam ab8895), H3K4me3 (Abcam ab8580), H3K27me3 (Millipore 07–449) AR (Millipore PG-21), BRD4 (Sigma HPA015055).

For ChIP-PCR analysis, the following primers were used: NR3C1 enhancer - F: ACCAGACTGAATGTGCAAGC; R: AGGGTTTTTGATGGCACTGA

For ChIP-Seq, libraries were made using the KAPA Biosystems Hyper Library Prep Kit (cat. # KK8504), using 10 ng of DNA as input and 10 PCR cycles for library amplification. The samples were done as single runs per sample on a HiSeq 2500, as rapid run v2 chemistry, single read 50.

Reads were aligned to the human genome (hg19) using Bowtie (v1.1.1) (*Langmead et al., 2009*) with default parameters and the results were converted to bam files by Samtools (*Li et al., 2009*). Reads mapped to a single genomic location were kept and redundant ones were filtered out. Tdf files were generated from the bam files by Igvtools with default parameters for visualization in IGV (*Robinson et al., 2011*).

For organoid ChIP-seq analysis, reads were aligned to the human genome (hg19) using BWA aln (v0.7.4) with default parameters. Reads mapped to more than two genomic loci were filtered out. Tdf files were generated from the bam files by IGV tools with default parameters for visualization in IGV.

## ChromHMM GR (NR3C1) locus analysis

Histone modification ChIP-seq data (H3K27ac, H3K4me1, H3K4me3) of VCaP were downloaded from the Gene Expression Omnibus (GEO) (GSM1328982, GSM353631 and GSM353620, respectively). The ChIP-seq data from brain, breast vHMEC, CD34/4/8 cells, colon smooth muscle, gastric, kidney, heart, liver, lung, pancreas, penis foreskin fibroblast primary cells, rectal smooth muscle, small intestine and stomach mucosa were downloaded from the Roadmap Epigenomics project (*Kundaje et al., 2015*), while the data for h1ESC, HEPG2, K562 and NHEK were from the ENCODE database (*ENCODE Project Consortium, 2012*). Unique reads mapped to a single genomic location were kept. ChromHMM (v1.12) was used to predict chromatin states with default parameters (*Ernst and Kellis, 2012*).

## CRISPR of GR (NR3C1) enhancer

CRISPR sgRNA guides were cloned into the PX458 plasmid vector (Addgene plasmid #48138) as previously described (*Ran et al., 2013*). sgRNA primer guide pairs flanking the GR (NR3C1) enhancer were as follows: sgRNA_1: CACCGTTAATTTCGCCCCCGTCCTG, sgRNA_2: CACCGAATTGTGACTATCAGAGGCT; sgRNA_3: CACCGAGGGGAGGGAATGTACGAAT, sgRNA_4: CACCGGAATTGTGACTATCAGAGGC

LREX'Enz cell lines were transiently transfected using Lipofectamine 2000 (Thermo Fisher Scientific) as per the standard protocol using the PX458 guide pairs, and sorted for GFP expression 4

days post-transfection. GFP-positive cells were then harvested for genomic DNA and RNA using DNEasy kit and RNEasy kit (Qiagen), respectively.

PCR primers used for amplifying and sequencing the excised enhancer were as follows: GRe_-flank_F- CACACAATCCCATTTTGCAG, GRe_flank_R - TAGCGCTCCCAGGCTTATTA, GR_internal_F - ACCAGACTGAATGTGCAAGC

## RT-qPCR analysis

RNA was harvested from cell lines and tumors using RNeasy kit (Qiagen). cDNA was generated using High Capacity cDNA Reverse Transcription Kit (Applied Biosystems). Data was always normalized relative to ACTB. Primers for GR, AR, SGK1, NKX3_1, and ACTB were all purchased through Qiagen.

## Analysis of GR expression and AR alterations across multiple datasets

Normalized RNA-seq expression values from datasets of The Cancer Genome Atlas (TCGA) (*Cancer Genome Atlas Research Network et al., 2013*) and PCF/SU2C (poly-A RNA-seq samples) (*Robinson et al., 2015*) were used. N3RC1 RNA expression was normalized by dividing its expression by the expression of the housekeeping gene UBC. Significantly different values were assessed using Wilcoxon rank sum test.

AR mutations and AR-V7 expression values were from the PCF/SU2C dataset. AR-V7 expression measured by normalized RNA-seq read counts across splice junction.

Tumor purity content was estimated computationally using the ABSOLUTE method (*Carter et al., 2012*), based on mutant allele variant fractions and zygosity shifts. Stromal signature score was applied to the normalized RNA-seq expression dataset (*Yoshihara et al., 2013*).

## In vivo xenograft experiments

Tumors xenograft experiments were performed as previously described (*Arora et al., 2013*). 2 million cells were injected subcutaneously into the flank of castrated CB17 SCID mice in a 50:50 mix of matrigel and regular culture medium. Measurements were obtained weekly using Peira TM900 system (Peira bvba, Belgium). All animal experiments were performed in compliance with the guidelines of the Research Animal Resource Center of Memorial Sloan Kettering Cancer Center. Drug treatments included: enzalutamide (10 mg/kg in vehicle 1% carboxymethyl cellulose, 0.1% Tween-80, 5% DMSO) 5 days a week by oral gavage; JQ1 (50 mg/kg in vehicle 10% hydroxypropyl beta cyclodextrin) 5 days a week by intraperitonial injection; doxycycline treatment provided through water and food. Tumors were started on treatment once they reached a sufficient size (200 mm$^3$), and harvested 4 weeks post-treatment start.

## Transcriptome analysis

RNA was harvested from cell lines and tumors using RNeasy kit (Qiagen). RNA sequencing libraries were prepared using the KAPA Hyper Prep Kit (kapabiosystems) in accordance with the manufacturer's instructions. Briefly, 40 ng (or less if not available) of total RNA was used for cDNA synthesis and amplification using the Ovation RNA-Seq System (Nugen). cDNA then was adenylated, ligated to Illumina sequencing adapters, and amplified by PCR (using 10 cycles). Final libraries were evaluated using fluorescent-based assays including PicoGreen (Life Technologies) or Qubit Fluorometer (invitrogen) and Fragment Analyzer (Advanced Analytics) or BioAnalyzer (Agilent 2100), and were sequenced on an Illumina HiSeq2500 sequencer (v4 chemistry) using $2 \times 50$ bp cycles.

Reads were aligned to the human genome (hg19) using STAR (v2.4.2a) (*Dobin et al., 2013*). The number of RNA-seq fragments mapped to each gene was determined for all genes in the GEN-CODE database (v19) using the HTSeq (*Harrow et al., 2012*; *Anders et al., 2015*). DESeq2 was used to determine differentially expressed genes at false discovery rate <0.05 (*Love et al., 2014*).

## Gene set enrichment analysis (GSEA)

GSEA analysis was performed from gene sets that were adapted from a method as previously described (*Arora et al., 2013*) using software publicly available from the Broad Institute (http://www.broadinstitute.org/gsea/index.jsp). The gene sets used were as follows: AR-biased gene set - *NKX3-1, PIK3AP1, TIPARP, PRAGMIN, DKFZP761P0423, ENDOD1, SLC2A3, C1ORF116, TMPRSS2, PAK1IP1, CROT, FZD5, GADD45G, ZNF385B, SLC36A1*; GR-biased gene set - *SGK1, TUBA3E,*

*SCNN1G, TUBA3C, DDIT4, EMP1, KRT80, TUBA3D, ACTA2, RGS2, C9ORF152, PNLIP, PPAP2A, SLC25A18, S100P, SPSB1, HSD11B2, LOC440040, SPRYD5, TRIM48, KLF9, PGC, LOC340970, ZNF812, PRR15L, PGLYRP2, BCL6, LOC399939, AZGP1, PRKCD, LOC100131392, GADD45B, ZBTB16, EEF2K, CRY2, LIN7B, KIAA0040, FKBP5, STK39, CGNL1, MT1X*

### Chem-Seq and analysis

Chem-seq was performed as previously described using biotinylated JQ1 compound (*Anders et al., 2014*).

Library preparation and sequencing was performed as described earlier, same as ChIP-seq protocol.

Reads were aligned to the human genome (hg19) using Bowtie. Non redundant reads mapped to a single genomic location were kept for peak calling by the MACS2 (v2.1.0) software (*Zhang et al., 2008*), with parameter –q 0.5. Peaks of differential binding were determined by DiffBind (*Ross-Innes et al., 2012*).

### Tumor xenograft and tissue microarray IHC

Immunohistochemsitry (IHC) was performed on tumor xenografts and tissue microarrays as previously described (*Arora et al., 2013*). Tumors were fixed in 4% PFA prior to paraffin embedding and then were stained for GR at 1:200 with anti-glucocorticoid receptor (D6H2L) XP Rabbit mAb (Cell Signaling Technology, #12041) using the Ventana BenchMark ULTRA.

### Statistics

All RT-qPCR and xenograft volume change comparisons are by two-sided t test. In meta-analysis across multiple large RNA-seq datasets, significantly different values were assessed using Wilcoxon rank sum test.

For in vivo tumor transcriptome analysis, DESeq2 was used to determine differentially expressed genes at false discovery rate <0.05. Four tumors were used per tumor condition.

GSEA statistical analysis was carried out with publicly available software from the Broad Institute (http://www.broadinstitute.org/gsea/index.jsp). In all figures, *p<0.05, **p=<0.01, ***p=<0.001, and ****p=<0.0001.

For Z-score ChIP-seq statistics, loci in figures were split into bins, and normalized read counts (per 10 million reads) were calculated for each bin. Z-scores were calculated to find bins with more ChIP-seq reads than the rest in each sample.

## Acknowledgements

We thank Adriana Heguy and her team at the NYU Genome Technology Center for their help with the ChIP-seq library prep and sequencing, Jay Bradner (Novartis) and Jun Qi (Dana Farber Cancer Institute) for generously providing JQ1 and bio-JQ1 and their help with the Chem-seq protocol, the New York Genome Center for conducting the RNA-sequencing, and Marina Asher at the MSKCC Pathology Core for assistance with IHC staining of patient samples.

## Additional information

#### Competing interests

Charles L Sawyers: Senior editor, *eLife*. John Wongvipat: JW is a co-inventor of enzalutamide and receive royalties from the University of California. The other authors declare that no competing interests exist.

#### Funding

| Funder | Grant reference number | Author |
|---|---|---|
| National Institutes of Health | R01CA193910 | Matthew L Freedman |
| Prostate Cancer Foundation | Young Investigator Award | Vivek K Arora |

| National Institutes of Health | R01CA155169 | Charles L Sawyers |
| National Institutes of Health | R01CA19387 | Charles L Sawyers |
| National Institutes of Health | P50CA092629 | Charles L Sawyers |
| Howard Hughes Medical Institute | | Charles L Sawyers |

The funders had no role in study design, data collection and interpretation, or the decision to submit the work for publication.

## Author contributions

Neel Shah, Conceptualization, Resources, Data curation, Formal analysis, Supervision, Investigation, Methodology, Writing—original draft, Writing—review and editing; Ping Wang, Data curation, Formal analysis, Investigation, Methodology, Writing—review and editing; John Wongvipat, Resources, Data curation, Investigation, Methodology; Wouter R Karthaus, Investigation, Methodology, Writing—review and editing; Wassim Abida, Conceptualization, Data curation, Formal analysis, Methodology; Joshua Armenia, Shira Rockowitz, Yotam Drier, Data curation, Formal analysis, Methodology; Bradley E Bernstein, Henry W Long, Matthew L Freedman, Conceptualization, Resources, Data curation, Formal analysis, Funding acquisition, Investigation, Methodology, Writing—review and editing; Vivek K Arora, Deyou Zheng, Conceptualization, Data curation, Formal analysis, Supervision, Funding acquisition, Investigation, Methodology, Writing—review and editing; Charles L Sawyers, Conceptualization, Resources, Funding acquisition, Investigation, Methodology, Writing—original draft, Writing—review and editing

## Author ORCIDs

Vivek K Arora [ID] http://orcid.org/0000-0003-1694-9109
Deyou Zheng [ID] http://orcid.org/0000-0003-4354-5337
Charles L Sawyers [ID] http://orcid.org/0000-0003-4955-6475

## Ethics

Animal experimentation: All animal experiments were performed in compliance with the guidelines of the Research Animal Resource Center of Memorial Sloan Kettering Cancer Center.

## Decision letter and Author response

Decision letter https://doi.org/10.7554/eLife.27861.018
Author response https://doi.org/10.7554/eLife.27861.019

# Additional files

## Supplementary files

• Transparent reporting form
DOI: https://doi.org/10.7554/eLife.27861.011

## Major datasets

The following dataset was generated:

| Author(s) | Year | Dataset title | Dataset URL | Database, license, and accessibility information |
| --- | --- | --- | --- | --- |
| Shah N, Wang P, Wongvipat J, Karthaus WR, Abida W, Armenia J, Rockowitz S, Drier Y, Bernstein BE, Long HW, Freedman ML, Arora VK, Zheng D, Sawyers CL | 2017 | Regulation of the glucocorticoid receptor via a BET-dependent enhancer drives antiandrogen resistance in prostate cancer | https://www.ncbi.nlm.nih.gov/geo/query/acc.cgi?acc=GSE103449 | Publicly available at the NCBI Gene Expression Omnibus (accession no. GSE103449) |

The following previously published datasets were used:

| Author(s) | Year | Dataset title | Dataset URL | Database, license, and accessibility information |
|---|---|---|---|---|
| Stand Up To Cancer East Coast Prostate Cancer Research Group | 2015 | Integrative clinical genomics of advanced prostate cancer | https://www.ncbi.nlm.nih.gov/gap/?term=phs000915.v1.p1 | Publicly available at NCBI dbGap (accesion no. phs000 915.v1.p1) |
| NIH Roadmap Epigenomics Project | 2010 | NIH Roadmap Epigenomics Project Data Listings | https://www.ncbi.nlm.nih.gov/geo/roadmap/epigenomics/ | Publicly available at the NCBI (https://www.ncbi.nlm.nih.gov/) |

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
