## [Decision Letter]

Thank you for submitting your article "A non-genomic mechanism of antiandrogen resistance mediated by BET-dependent expression of the glucocorticoid receptor" for consideration by *eLife*. Your article has been favorably evaluated by Sean Morrison (Senior Editor) and three reviewers, one of whom, Maarten van Lohuizen (Reviewer #1), is a member of our Board of Reviewing Editors. The following individual involved in review of your submission has agreed to reveal their identity: Luke Gaughan (Reviewer #3).

The reviewers have discussed the reviews with one another and the Reviewing Editor has drafted this decision to help you prepare a revised submission. While they all deem the work very interesting, several concerns have been raised that need to be addressed in a revision.

Major points:

1) One question that stands out is how clinically relevant/prevalent this new resistance mechanism is? Can the authors provide any indication of how frequently this may occur in patients, as opposed to well-known other paths to resistance, such as selection for mutations in AR signaling? How relevant is this enhancer element in patients who are resistant to Enzalutamide via AR gene mutation or generation of splice mutants? This is especially relevant considering that while the results presented are very significant for the LNAR/LREX cell system used, the effects seen in TCGA patient data appear to be much weaker (Figure 2: minor increase in GR expression and huge inter-patient variability between CRPC pre- treatment versus post treatment). Have the authors been able to test the existence of this resistance mechanism also on CRPC patient-derived tumor organoids?

2) Figure 2: Interesting how overall, the expression of GR in CRPC + Enzalutamide is still markedly lower than normal and primary material. Why do you think this is?

Does Figure 2 take stromal versus luminal components into consideration? This is important since GR is strongly found in stroma (See Figure 2—figure supplement 1). Also, comparison between primary and post-enzalutamide CRPC would further support the claims. Please include the number of samples studied in the figure. When possible, paired statistics between pre-post enzalutamide would be ideal.

3) Figure 3 and Figure 4: GR levels are sensitive to JQ-1 in LREX cells. Does JQ-1 enhance AR association at the enhancer element? Does the combination of JQ-1 and EZH2 inhibitor re-activate GR expression? The mechanism of how BET inhibition regulates GR expression is a little unclear at the level of chromatin. The RNA-Seq data is interesting, but how the factors that govern H3K27methylation, H3K27 acetylation, and involvement of BET proteins needs to be more explicit. ChIP experiments in LREX cells in the presence and absence of Enzalutamide +/- JQ-1 to assess all BET family members and impact on proximal and enhancer acetylation and methylation marks would be really informative. In addition, In the description of Figure 3, the authors claim no effect of JQ1 on AR-responsive genes, such as NKX3.1 However, this statement needs further analyses, since only 1 example gene cannot rule out effects on total AR. This is particularly relevant, since the authors state that JQ1 indeed affects AR target genes (Asangani et al., 2014; Faivre et al., 2017; Chan et al., 2015).

Other points:

4) Figure 1 shows the generation and use of LREX' cells, by in vivo exposure of LNCAP/AR cells to Enzalutamide. How many clones were generated, and how consistent are these findings over the different clones? Please include this information in the paper.

5) Figure 1, Figure 2, Figure 4: please quantify reads in the enhancer peak and other regions, provide statistics and normalize over library depth. Quantifiable statements are made, but this can only be done after quantifying the signal.

6) Figure 2: If technically possible, re-ChIP experiments to study interaction between AR and EED/SUZ12/EZH2 would be interesting to conduct to help confirm the involvement of the repressive PRC complex. The AR ChIP-seq data suggests loss of AR binding at the enhancer upon Enzalutamide treatment. Would you expect less steady-state AR binding to this site in the LREX cells versus LNAR cells given higher overall levels of GR in the LREX cells (as shown in Arora et al)?

7) Figure 2—figure supplement 1 and E lack error bars.

8) In the Discussion section, the authors hypothesize that genomic selectivity of bio-JQ1 may be due to differences in H3K27ac. However, since the authors already generated H3K27ac data (Figure 1), these analyses could readily be done, hopefully confirming this hypothesis.

9) No information is provided on how many replicates for ChIP-seq and Chem-seq were performed. Please provide this information.

10) The title appears not well chosen. In the end, the mechanism is genomic. Please provide a better, more succinct, title, keeping in mind the *eLife* instructions for titles.

---

## [Author Response]

*Major points:*

*1) One question that stands out is how clinically relevant/prevalent this new resistance mechanism is? Can the authors provide any indication of how frequently this may occur in patients, as opposed to well-known other paths to resistance, such as selection for mutations in AR signaling? How relevant is this enhancer element in patients who are resistant to Enzalutamide via AR gene mutation or generation of splice mutants? This is especially relevant considering that while the results presented are very significant for the LNAR/LREX cell system used, the effects seen in TCGA patient data appear to be much weaker (Figure 2: minor increase in GR expression and huge inter-patient variability between CRPC pre- treatment versus post treatment). Have the authors been able to test the existence of this resistance mechanism also on CRPC patient-derived tumor organoids?*

The clinical relevance of GR as a resistance mechanism in castration resistant prostate cancer (CRPC) is a critical issue that remains under active investigation in the prostate cancer community, particularly since clinical trials of GR antagonists and BET inhibitors have been launched based on the work of several groups. We address the specific points below:

First, at least three independent groups have now published comprehensive manuscripts implicating GR as a resistance mechanism in preclinical models of CRPC and in patients (Arora VK, et al. Cell, 2013; Isikbay M, et al. Horm Cancer, 2014; Li J, et al. *eLife*, 2017). In addition, ASCO abstracts from two leading cancer centers have documented GR upregulation in circulating tumor cells (CTCs) of CRPC patients and in surgical samples from patients with high risk localized disease receiving neoadjuvant therapy (Wise D, et al. ASCO abstract 122964, 2016; Efstathiou E, et al. ASCO abstract 109810, 2015). These references are now cited in the revised manuscript.

Second, we have more than doubled the number of patients analyzed in Figure 2 of the original submission (from 19 patients to 50 patients), which now includes patients who have progressed on either enzalutamide (Enz) or abiraterone (Abi). GR levels remain significantly higher in the post-Enz/Abi patients (revised Figure 2). As before, there is considerable inter-patient variability which we address further in our response to major point 2.

Third, to address whether tumors with GR expression also have other, well documented resistance mechanisms, we analyzed the 50 post Enz/Abi tumors for AR mutations, AR splice variants (AR-V7) and total AR mRNA levels. 7 of the 50 patients had AR mutations, all with AR L702H mutation which allows AR to be activated by glucocorticoids and has been reported in patients post-Abi (Abi must be given in combination with prednisone to prevent endocrine side effects from CYP17 inhibition). There is no correlation of this AR mutation with GR levels, although numbers are small. This data is presented in new Figure 2—figure supplement 1. Interestingly, AR-V7 levels are negatively correlated with GR expression in this cohort, suggesting that GR upregulation and AR-V7 expression may be mutually exclusive in post-Enz/Abi patients. This data is presented in new Figure 2—figure supplement 1.

Fourth, we have now documented that GR activation confers Enz resistance in a patient-derived, Enz-sensitive human prostate cancer organoid model (MSK-PCa2) by showing that treatment with the GR agonist dexamethasone (Dex) partially rescues growth and target gene expression with Enz treatment, but Dex alone does not have any growth effect. This data is presented in new Figure 1—figure supplement 1.

In summary, we have addressed the clinical relevance question by including data from 31 additional CRPC patients and from new patient-derived organoids, all of which provides further support for the GR hypothesis. Going forward, the best test of the clinical relevance question will emerge from various ongoing clinical trials.

*2) Figure 2: Interesting how overall, the expression of GR in CRPC + Enzalutamide is still markedly lower than normal and primary material. Why do you think this is?*

*Does Figure 2 take stromal versus luminal components into consideration? This is important since GR is strongly found in stroma (See Figure 2—figure supplement 1). Also, comparison between primary and post-enzalutamide CRPC would further support the claims. Please include the number of samples studied in the figure. When possible, paired statistics between pre-post enzalutamide would be ideal.*

This is a great question. As suggested by the referees, we have considered the possibility that differences in tumor versus stromal content in the pre- versus post-Enz/Abi samples could explain the difference in GR expression. Tumor purity content was estimated computationally using the ABSOLUTE method (Carter SL, et al. Nat Biotechnol., 2012), based on mutant allele variant fractions and zygosity shifts. We observed no significant differences in the pre- versus post-Enz/Abi settings. We also applied a stromal signature score (Yoshihara K, et al. Nat Comm, 2013) and again saw no differences. These analyses are presented in new Figure 2—figure supplement 1. We interpret these results as evidence that the increase in GR in the post-Enz/Abi samples is most likely explained by expression in tumor cells.

We have also considered another possibility, based on the fact that our model predicts that GR levels should be highest in patients who remain on Enz at time of progression. [We have shown in this manuscript that AR binds directly to the GR enhancer to inhibit its transcription and that GR levels drop when Enz is removed.] Because Enz or Abi is typically discontinued in patients at the time of disease progression, many patients annotated as “post-Enz/Abi” in this dataset may have no longer been on drug at the time of the biopsy. To explore this, we went back to the clinical records to determine the date that the biopsies were taken for RNA-seq relative to the time of Enz/Abi discontinuation and found that, indeed, the majority of patients were already off drug. We then asked if there was a relationship between GR levels and time off drug, using 40 days as a cut point for this analysis. Interestingly, GR levels are higher in patients whose biopsy was obtained within 40 days compared to those beyond 40 days, although the difference does not meet statistical significance (p=0.0932) (see figure below, for referees). Based on this trend, we suspect that GR levels might be significantly higher if biopsies were taken at time of progression while patients were still on drug. We have elected not to include this analysis in the revision since the results are not conclusive but have added a sentence in the Discussion.

For the query about sample numbers and paired statistics, we have included the sample number data in revised Figure 2 but we are unable to perform paired statistics due to the very small number of paired samples. Collection of matched pairs is a top priority in our ongoing Prostate Cancer Dream team collaboration.

**Author response image 1. respfig1:** Top – Clinical data for post Enz/Abi patients. Arrows indicate the time at which the biopsy was taken for sequencing (number of days before/after Enz/Abi stop date). Bottom – Patients whose biopsies were taken while off drug for less than 40 days had a trend for higher GR expression when compared to patients who were off drug for greater than 40 days (p-val=0.0932).

*3) Figure 3 and Figure 4: GR levels are sensitive to JQ-1 in LREX cells. Does JQ-1 enhance AR association at the enhancer element? Does the combination of JQ-1 and EZH2 inhibitor re-activate GR expression? The mechanism of how BET inhibition regulates GR expression is a little unclear at the level of chromatin. The RNA-Seq data is interesting, but how the factors that govern H3K27methylation, H3K27 acetylation, and involvement of BET proteins needs to be more explicit. ChIP experiments in LREX cells in the presence and absence of Enzalutamide +/- JQ-1 to assess all BET family members and impact on proximal and enhancer acetylation and methylation marks would be really informative. In addition, In the description of Figure 3, the authors claim no effect of JQ1 on AR-responsive genes, such as NKX3.1 However, this statement needs further analyses, since only 1 example gene cannot rule out effects on total AR. This is particularly relevant, since the authors state that JQ1 indeed affects AR target genes (Asangani et al., 2014; Faivre et al., 2017; Chan et al., 2015).*

These are a series of important questions about the mechanism of JQ1 activity in the LREX model and potential connections with EZH2. We have included a series of additional ChIP-PCR experiments (for AR, H3K27ac, H3K4me1, H3K27me3, in triplicate) in LREX’ cells +/- Enz to address the query about AR binding and histone marks. JQ1 treatment does not enhance AR binding at the GR enhancer (new Figure 3—figure supplement 1). Further, JQ1 treatment does not alter histone modification levels at the GR locus, consistent with the prevailing hypothesis that JQ1 primarily targets epigenetic readers without altering the activity of writers or erasers.

As for the query about combined EZH2+BET inhibition, we found that GSK126 (EZH2 inhibitor) + JQ1 does not re-activate GR expression (see Author response image 2), which we believe is expected since we do not see an increase in H3K27me3 levels with JQ1 treatment.

**Author response image 2. respfig2:** LREX' cells treated with different combinations of 1uM JQ1 and 3uM GSK126 for 72h. Combination of JQ1+GSK126 does not rescue GR expression.

Regarding the request for ChIP experiments with all BET family members, we were only able to generate reliable data for BRD4 despite multiple attempts with antibodies to various other BET family members (particularly BRD2). We learned from conversations with several other colleagues that ChIP experiments with these other BET proteins can be particularly challenging. Indeed, this is precisely the reason we turned to JQ1 Chem-Seq, which serves as a surrogate for binding of all BET bromodomain proteins across the genome (Figure 4).

The question about NKX3.1 and other AR targets led us to examine a panel of AR/GR target genes that we selected based on our previous annotation as AR-biased (NKX3.1, TMPRSS2) versus GR-biased (SGK1, FKBP5) versus both (PSA), based on RNAseq analysis after DHT or Dex treatment (Arora et al. Cell, 2013). In a JQ1 dose titration experiment, we found that GRbiased genes, as well as GR itself and MYC, were inhibited in a dose-dependent fashion whereas AR-biased genes (NKX3.1, TMPRSS2) were paradoxically activated at low doses (<100nM) and only modestly inhibited at high doses (new Figure 3). We are quite excited by this result because it offers a compelling explanation for the relatively GR-selective effects on JQ1 we observe in vivo (due to the suboptimal in vivo pharmacological profile of JQ1 in mouse models) and suggests there may be a favorable therapeutic index for BET inhibition in CRPC patients whose tumors have increased GR expression. These data are now incorporated into the revised manuscript.

*Other points:*

*4) Figure 1 shows the generation and use of LREX' cells, by* in vivo *exposure of LNCAP/AR cells to Enzalutamide. How many clones were generated, and how consistent are these findings over the different clones? Please include this information in the paper.*

8 of 13 Enz-resistant tumor clones showed GR upregulation at varying levels (Arora VK, et al. Cell, 2013). We thawed three clones (two GR-positive and one GR-negative) and conducted flow cytometry analysis analogous to that done for LREX’. Consistent with the LREX' clone, GR levels were reversible upon Enz exposure in the 2 GR-positive clones. However, the GRnegative clone showed no GR induction in the presence or absence of Enz, consistent with a different mechanism of Enz-resistance. These data are presented in new figure Figure 1—figure supplement 1.

*5) Figure 1, Figure 2, Figure 4: please quantify reads in the enhancer peak and other regions, provide statistics and normalize over library depth. Quantifiable statements are made, but this can only be done after quantifying the signal.*

Quantified normalized reads and statistics were added to all figure legends.

*6) Figure 2: If technically possible, re-ChIP experiments to study interaction between AR and EED/SUZ12/EZH2 would be interesting to conduct to help confirm the involvement of the repressive PRC complex. The AR ChIP-seq data suggests loss of AR binding at the enhancer upon Enzalutamide treatment. Would you expect less steady-state AR binding to this site in the LREX cells versus LNAR cells given higher overall levels of GR in the LREX cells (as shown in Arora et al)?*

Unfortunately, we were unable to obtain reliable results through re-ChIP experiments. However, we were able to explore the potential interaction between AR and EZH2, in part, by performing EZH2 ChIP-PCR at the GR locus in the presence or absence of Enz treatment, using the MYT1 promoter as a positive control. We see a clear reduction in EZH2 binding in LREX’ cells versus LNAR' cells, consistent with the decrease in H3K27me3 at this locus. However, EZH2 binding is unchanged upon Enz treatment, suggesting that AR and EZH2 are not directly interacting since Enz displaces AR from the GR enhancer. Because the results are negative, we have not added this data to the revision but can certainly do so if requested by the referees.

**Author response image 3. respfig3:** EZH2 ChIP-PCR performed in LNAR' and LREX' cells with or without Enz treatment at GR promoter (top), and MYT1 promoter (bottom) used as a positive control.

We are unclear about the referee’s point in asking about steady-state AR binding but considered the possibility that, in LREX’ cells, GR might compete with AR for binding at the enhancer. However, steady-state AR binding at this site is comparable in LREX vs LNAR cells (Figure 2 in the revision). Interestingly, we did find that DHT treatment of LREX’ cells, which promotes AR binding, can reduce GR binding at the NR3C1 locus (where AR and GR can both bind), suggesting that there is likely competition for binding at certain sites (see Author response image 4).

**Author response image 4. respfig4:** GR ChIP-seq peaks at NR3C1 locus for LREX'Enz samples. Cells were treated with either Dex (100nM 24h – top track) or Dex + DHT (1nM 24h – bottom track). Acute treatment with DHT is able to significantly decrease the peak height of GR at this binding site.

*7) Figure 2—figure supplement 1 and E lack error bars.*

Thank you for noting this. Error bars have been added.

*8) In the Discussion section, the authors hypothesize that genomic selectivity of bio-JQ1 may be due to differences in H3K27ac. However, since the authors already generated H3K27ac data (Figure 1), these analyses could readily be done, hopefully confirming this hypothesis.*

Thanks for this suggestion. We see a significant positive correlation when we plot bioJQ1 selectivity vs. H3K27ac peak changes, confirming the hypothesis. This new data is presented in Figure 4—figure supplement 1.

*9) No information is provided on how many replicates for ChIP-seq and Chem-seq were performed. Please provide this information.*

This information was added to Materials and methods. ChIP-seq was performed as a single run per sample on a HiSeq 2500. Peaks of interest were validated via ChIP-PCR in triplicate.

*10) The title appears not well chosen. In the end, the mechanism is genomic. Please provide a better, more succinct, title, keeping in mind the eLife instructions for titles.*

The title was changed accordingly: "Regulation of the glucocorticoid receptor via a BETdependent enhancer drives anti-androgen resistance in prostate cancer"